# The Role of Daily Adaptive Stereotactic MR-Guided Radiotherapy for Renal Cell Cancer

**DOI:** 10.3390/cancers12102763

**Published:** 2020-09-25

**Authors:** Shyama U. Tetar, Omar Bohoudi, Suresh Senan, Miguel A. Palacios, Swie S. Oei, Antoinet M. van der Wel, Berend J. Slotman, R. Jeroen A. van Moorselaar, Frank J. Lagerwaard, Anna M. E. Bruynzeel

**Affiliations:** 1Department of Radiation Oncology, Amsterdam University Medical Centers, 1081 HZ Amsterdam, The Netherlands; su.tetar@amsterdamumc.nl (S.U.T.); o.bohoudi@amsterdamumc.nl (O.B.); s.senan@amsterdamumc.nl (S.S.); m.palacios@amsterdamumc.nl (M.A.P.); ss.oei@amsterdamumc.nl (S.S.O.); a.vanderwel1@amsterdamumc.nl (A.M.v.d.W.); bj.slotman@amsterdamumc.nl (B.J.S.); fj.lagerwaard@amsterdamumc.nl (F.J.L.); 2Department of Urology, Amsterdam University Medical Centers, 1081 HV Amsterdam, The Netherlands; rja.vanmoorselaar@amsterdamumc.nl

**Keywords:** MR-guided, radiotherapy, MRgRT, stereotactic ablative radiotherapy, stereotactic ablative radiation therapy (SABR), renal cell cancer, RCC, online adaptive

## Abstract

**Simple Summary:**

Standard treatment for localized renal cell carcinoma (RCC) is surgery. Stereotactic radiotherapy given in a few high dose fractions is a promising treatment for this indication and could be an alternative option for patients unsuitable for surgery. Stereotactic MR-guided radiotherapy (MRgRT) is clinically implemented as a new technique for precise treatment delivery of abdominal tumors, like RCC. In this study, we evaluated the clinical impact of stereotactic MRgRT given in five fractions of 8 Gy and routine plan re-optimization for 36 patients with large primary RCCs. Our evaluation showed good oncological results with minimal side-effects. Even in this group with large tumors, daily plan re-optimization was only needed in a minority of patients who can be identified upfront. This is a favorable result since online MRgRT plan adaptation is a time-consuming procedure. In these patients, MRgRT delivery will be faster, and these patients could be candidates for even less fractions per treatment.

**Abstract:**

Novel magnetic-resonance-guided radiotherapy (MRgRT) permits real-time soft-tissue visualization, respiratory-gated delivery with minimal safety margins, and time-consuming daily plan re-optimisation. We report on early clinical outcomes of MRgRT and routine plan re-optimization for large primary renal cell cancer (RCC). Thirty-six patients were treated with MRgRT in 40 Gy/5 fractions. Prior to each fraction, re-contouring of tumor and normal organs on a pretreatment MR-scan allowed daily plan re-optimization. Treatment-induced toxicity and radiological responses were scored, which was followed by an offline analysis to evaluate the need for such daily re-optimization in 180 fractions. Mean age and tumor diameter were 78.1 years and 5.6 cm, respectively. All patients completed MRgRT with an average fraction duration of 45 min. Local control (LC) and overall survival rates at one year were 95.2% and 91.2%. No grade ≥3 toxicity was reported. Plans without re-optimization met institutional radiotherapy constraints in 83.9% of 180 fractions. Thus, daily plan re-optimization was required for only a minority of patients, who can be identified upfront by a higher volume of normal organs receiving 25 Gy in baseline plans. In conclusion, stereotactic MRgRT for large primary RCC showed low toxicity and high LC, while daily plan re-optimization was required only in a minority of patients.

## 1. Introduction

A radical or partial nephrectomy is the preferred standard curative treatment for localized renal cell carcinoma (RCC) [1,2,3,4]. Ablative local treatment, such as radiofrequency ablation (RFA), cryoablation (CA), or microwave ablation (MWA), is an alternative in elderly patients who present with a high surgical risk due to several comorbidities [3]. Radiotherapy does not have a prominent role in current international and national guidelines in treating primary RCC [1,2,3,4]. In recent years, stereotactic ablative radiation therapy (SABR) has been evaluated in several smaller retrospective and prospective studies [5,6,7,8,9,10,11,12,13,14], usually in RCC patients unsuitable for surgery. Outcomes of a multi-institutional pool from nine institutions, utilizing either single or multi-fractionated treatment in 223 patients, have been reported by the International Radiosurgery Oncology Consortium for Kidney (IROCK) [15]. SABR for RCC was found to be well tolerated, achieved local control (LC) rates exceeding 95% at four years of follow-up and grade ≥3 toxicity rates of 1.3%, and had an average decrease in glomerular filtration rate of 5.5 mL per minute. The majority of the tumors in this pooled analysis was ≤4 cm and clinical data for larger tumors is limited. A retrospective analysis of a subgroup of 95 patients with tumors >4 cm was recently published [16], but with the exception of these data, clinical outcomes on cT1b-T2 RCC SABR are scarce. Due to the inherent limitations to a pooled analyses, the Trans-Tasman Radiation Oncology Group (TROG) and the Australian and New Zealand Urogenital and Prostate Cancer Trials Group (ANZUP) have initiated a prospective, multi-institutional phase II study in 70 patients with biopsy-confirmed medical inoperable RCC patients [17]. Full accrual has recently been completed, and the data of this trial are eagerly awaited.

Technical challenges in renal SABR include the management of intra-fractional motion, and potential solutions using an internal target volume-approach, fiducial-assisted robotic SABR or abdominal compression [18] have been described. Magnetic-resonance (MR)-guided radiotherapy (MRgRT) has been considered a promising option because of its improved visualization of kidney tumors in relation to critical adjacent organs such as a small bowel, duodenum, and stomach and the opportunity of real-time tumor tracking and automated gated delivery [18,19]. MRgRT also facilitates daily plan re-optimization as a means to reduce organs at risk (OAR) doses when abdominal organs are near the primary tumor. Furthermore, MRgRT is an outpatient treatment for which no invasive procedures or anesthesia is required. However, to the best of our knowledge, clinical data on MR-guided SABR for localized RCC have not been reported.

Stereotactic MRgRT with routine daily plan adaptation was clinically implemented at our center in 2016 for a variety of clinical indications. The aim of the current paper is to describe our technique, early clinical outcomes, and the role of daily plan adaptation in MRgRT for patients with primary large RCC.

## 2. Materials and Methods

Data from all patients treated with MRgRT on the MRIdian-system (ViewRay Inc., Mountain View, CA, USA) at the Amsterdam University Medical Centers are collected within a prospective institutional review board approved database. Between May 2016 and February 2020, a total of 51 patients were treated for a primary RCC (*n* = 36), local recurrences (*n* = 5), renal metastases from other primary tumors (*n* = 3), or a diagnosis of urothelial carcinoma (*n* = 7). This analysis is restricted to the remaining 36 patients who were treated for primary RCC.

All patients underwent stereotactic adaptive MRgRT delivered to a dose of 40 Gy in five fractions in a two-week period. Implanted fiducials were not required, and the adaptive workflow was similar to that which had been described previously for pancreatic tumors [20]. Briefly, for simulation, both a MR-scan (0.35T True-FISP, TR/TE: 3.37 ms/1.45 ms, FA: 60°, 17-s with 1.6 mm × 1.6 mm × 3.0 mm resolution) and computed tomography (CT)-scan (slice thickness of 2 mm) are acquired during a shallow-inspiration breath-hold. Geometric accuracy of the MRIdian system is < 0.1 cm in a sphere of 10 cm radius around the isocenter, and <0.15 cm in a sphere of 17.5 cm radius. Every patient was brought as close to the isocenter as possible for each fraction, and the maximum distance from the tumor or any other critical structure to the isocenter was always below 10 cm. Geometric accuracy was assessed with two different dedicated phantoms for spatial integrity measurements. Contouring of the primary tumor (also called gross tumor volume; GTV) and OAR is performed on breath-hold MR-images with the aid of diagnostic imaging, generally contrast-enhanced CT scans. The PTV (planning target volume) is derived from the GTV plus an isotropic 3-mm margin. A co-planar baseline plan consisting of between 30 and 42 intensity modulated radiotherapy (IMRT)-segments is generated, using the MRIdian treatment planning software. Dose calculation was executed with a VMC and EGSnrc code-based Monte-Carlo algorithm (statistical uncertainty of 1% and a grid size of 0.3 cm × 0.3 cm × 0.3 cm) using the deformed electron density map from the simulation CT scan. Institutional target coverage and OAR constraints are summarized in Table 1.

We perform routine plan re-optimization using the daily pre-SABR breath-hold MR-imaging acquired in the treatment position. After rigid registration on the GTV, OAR contours are propagated to the repeat MR using deformable image registration. The ViewRay deformable image registration algorithm uses an intensity-based algorithm, which minimizes a cost function that measures the similarity between the images including a regularization term in order to obtain smoother deformation fields and prevent sharp discontinuities. The GTV and OAR contours are checked and adjusted where needed within a 2-cm distance of the PTV by the attending radiation oncologist. Next, the baseline IMRT plan is recalculated on the new anatomy (“predicted plan”), and subsequently re-optimized using the target and OAR optimization objectives of the baseline plan (“re-optimized plan”). Plan re-optimization prioritizes avoiding high doses to OARs, even when this is at the cost of decreased PTV coverage. Both the predicted and re-optimized plans are reviewed, and the re-optimized plan is selected for the actual delivery.

MRgRT delivery is performed using respiratory gating during subsequent breath-hold periods in shallow inspiration. The tracking structure for gating is either the primary tumor, or the kidney itself on a single sagittal plane (Figure 1), depending on the visibility on this sagittal plane. Gating is augmented by visual and/or auditory feedback provided to patients during treatment [21]. Visual feedback is performed with the aid of an in-room MR compatible monitor on which both the tracking structure (GTV or kidney) and the gating boundary (3 mm), generally corresponding to the PTV, is projected in real-time. The 2D MR images during treatment were acquired with a True FISP sequence with the MRIdian (0.35 T) at a frequency of four frames-per-second (TR: 2.1 ms, TE: 0.91 ms, FA: 60°). FOV was 0.35 cm × 0.35 cm and the slice thickness was 0.7 cm. Due to the low magnetic field and low FA, “real-time” MR images of the patient were performed without interruption during the beam-on time. A previous analysis showed a treatment duty cycle efficiency between 67% and 87% for upper abdominal tumors [22].

Baseline patient and tumor characteristics and follow-up data including LC, renal function, and toxicity were collected. Acute and late toxicity was scored using the Common Terminology Criteria for Adverse Events (CTCAE) version 4.0. Follow-up imaging was assessed by a CT-scan or ultrasound, and the tumor response was classified according to RECIST 1.1. criteria.

An offline analysis was performed to evaluate the need for daily plan re-optimization in MRgRT for RCC in a total of 180 fractions. For this purpose, predicted and re-optimized plans were analyzed for adherence with planning target objectives and OAR constraints, i.e., a V_38Gy_ of the GTV ≥ 90%, and V_33Gy_ ≤ 1 cc for stomach, duodenum, and bowel. Re-optimization was defined as “needed” when the predicted plan violated the above-mentioned GTV and/or OAR constraints, which was subsequently corrected by re-optimization. In contrast, plan re-optimization was defined as “redundant” when predicted plans already complied with the planning objectives. In addition, the value of plan re-optimization was analyzed on a patient level by studying the number of fractions per patient that were considered suboptimal.

### Statistical Analysis

Descriptive statistics were used for baseline patient and tumor characteristics. The change in renal function (eGFR) from baseline versus post-treatment at the latest available time point in follow-up was evaluated using the paired sampled *t*-test. Local, regional, distant disease control and overall survival (OS) were estimated using the Kaplan-Meier method. OS was calculated as the time between the first fraction of MRgRT and the date of the last follow-up. LC was calculated as the time between the first fraction of MRgRT and the date of last imaging. Statistical analysis used for plan comparisons was performed using the Wilcoxon Signed-Rank test. A *p*-value of < 0.05 was considered to be statistically significant. Decision tree analysis (CHAID, Chi-square automatic interaction detection) was used to explore predictive pretreatment characteristics and most significant cut-off values to identify patients for whom daily re-optimization was needed. Baseline volumetric, geometric, and dosimetric parameters, i.e., GTV size (cc), laterality (left, right), location (interpolar, upper or lower pole), V_33Gy_, V_30Gy_, V_25Gy_, and V_20Gy_ for each OAR structure separately or combined in one structure were used as input variables. The qualitative re-optimization benefit variable (“redundant” or “needed”) was selected as the target variable for decision tree analysis. The significance level for node splitting was set at *p* < 0.05. Stopping parameters to prevent over-fitting were applied by setting the minimum number of records in a leaf to be at least 10% of the data set. The Statistical Package for the Social Sciences (SPSS) version 26 (IBM^®^ SPSS Statistics, Armonk, NY, USA) was used to perform all statistical analyses.

## 3. Results

### 3.1. Clinical Outcomes

All 36 patients were referred for SABR after discussion in a multidisciplinary tumor board, and reasons for referral included a high surgical risk due to comorbidity (*n* = 9), which is unsuitable for other ablative therapies due to tumor size (*n* = 10) or location (*n* = 5), patient preference (*n* = 5), co-existing second malignancy (*n* = 3), use of anti-coagulants (*n* = 2), and chronic stage ≥IV kidney disease (*n* = 2). Baseline patient characteristics are summarized in Table 2. The mean age of this cohort was 78.1 years with a preponderance of men (66.7%). The mean tumor diameter was 5.6 cm (range 2.4–9.3 cm) with 86.1% of tumors measuring ≥4 cm in the largest dimension of which 23 patients have a cT1b tumor and 8 patients have a cT2a tumor. Five patients (13.9%) had metastasized renal cell carcinoma (RCC) at the time of diagnosis. Pathologic confirmation of RCC before treatment was achieved in approximately half of patients (55.6%) of which the majority was diagnosed with Fuhrman grade 2 (*n* = 14). Other patients with histology included Fuhrman grade 1 (*n* = 1), Fuhrman grade 3 (*n* = 1), a RCC with sarcomatoid features (*n* = 1), and a chromophobe tumor (*n* = 1). In two patients, no grading was available because pathologic confirmation was obtained from systemic metastases. All patients were able to complete adaptive MRgRT with an average time per fraction of 45 min. An overview of the average duration of the different components of adaptive MRgRT for RCC is shown in Figure 2. Three patients completed treatment while tracking on the kidney instead of the tumor.

The median follow-up was 16.4 months. Overall survival was 91.2% at one year (Figure 3), LC was 95.2% (Figure 3), and freedom from any progression was 91% at one year. Two patients had local recurrences. One patient had progressive distant disease at recurrence for which systemic therapy was delivered, and the second patient with an isolated local recurrence underwent radiofrequency ablation as salvage. Treatment-related acute toxicity grade ≥ 2 in the form of nausea was observed in a single patient, which responded to oral ondansetron. No other acute or late grade ≥2 toxicity was reported. The mean eGFR at baseline was 55.3 (SD ±19.0) mL/min/1.73 m^2^. With a mean interval of 16 months and mean eGFR post-MRgRT was 49.3 (SD ± 19.1) mL/min/1.73 m^2^, which indicates a decrease of 6.0 mL/min/1.73 m^2^. No patient in this cohort required dialysis during follow-up.

### 3.2. The Need for Daily Plan Re-Optimization

In 151 out of 180 fractions (83.9%), the predicted plans (without re-optimization) met all institutional target and OAR constraints. In these fractions, predicted and re-optimized plans were of similar quality with a mean GTV V_38Gy_ of 98.8% and 99.1%, respectively, and mean V_33Gy_ of 0 cc for both stomach, duodenum, and bowel. In the other 29 fractions, predicted plans were suboptimal with insufficient GTV coverage in two out of 180 fractions (1.1%) exceeding OAR constraints in 25 fractions (13.9%), and both insufficient GTV coverage and exceeded OAR constraints in another two fractions (1.1%). There was no significant difference in suboptimal predicted plans for left-sided or right-sided RCC (*p* = 0.56). For these suboptimal plans, on-couch re-optimization corrected the GTV V_38Gy_ from a mean of 88.7% (predicted) to 97.4% (re-optimized). Similarly, re-optimization corrected OAR V_33Gy_ ≤ 1 cc violations from on average V_33Gy_ of 4.1 (predicted plans) to 0.3 cc (re-optimized plans). Analysis on a patient basis showed that the 29 insufficient predicted fractions were distributed among 11 patients (11/36, 30.6%). However, three or more suboptimal fractions were seen in only five patients (13.9%).

Decision tree analysis identified the baseline OAR V_25Gy_ (combined structure of stomach, bowel, and duodenum) as the most significant predictor variable for daily adaptive planning needs with 0.5 cc as an optimal cut-off value (*p* < 0.001). In all cases with a baseline OAR V_25Gy_ of ≤ 0.5 cc, plan adaptation was redundant as the predicted plans already complied with institutional constraints. In patients with baseline OAR V_25Gy_ of more than 0.5 cc, plan re-optimization was needed in 32.2% of fractions in order to fulfill the preset target coverage and OAR constraints (Table 3). The correct classification rate of the decision tree was 86.1% with a sensitivity of 100% and a specificity of 67.7%. The difference between re-optimized and predicted dose parameters for target (GTV V_95%_) and OAR (V_33Gy_) stratified for split group 1 and 2 (Table 3) is shown in Figure 4.

## 4. Discussion

To the best of our knowledge, this is the first series of patients treated for primary RCC using MRgRT with routine daily plan re-optimization. We applied a commonly used fractionation scheme of 40 Gy in five fractions [18,23,24] in an overall treatment time of two weeks. Only a single patient reported nausea as acute toxicity, and no grade ≥ 2 late toxicity was observed. Despite the inclusion of large tumors, mostly T1b and T2, which had a mean tumor diameter of 5.6 cm and were generally unsuitable for other local therapies, we observed an LC rate of 95.2%. Our local response scoring has been according to the RECIST 1.1 criteria, and 83.3% had stable disease. In addition, 11.1% had partial remission, while 5.6% showed local progression. Fast tumor size regression is uncommon after SABR as previously reported by Sun and colleagues [11]. This preponderance of stable disease is in accordance with their paper. Both LC and OS are reported to be poorer for larger primary RCC than for the smaller lesions [25,26]. Despite this observation, our LC rate is within the high range of what was reported in recent systematic reviews, meta-analyses, and pooled analyses of SABR for primary RCC [15,24,27].

MRgRT with daily plan re-optimization was feasible with an average fraction duration of 45 min, even in poorer condition patients with multiple co-existing diseases. Despite this prolonged treatment duration, all patients were able to complete treatment, which indicates good tolerability. Our fractionation scheme of 40 Gy in five fractions is commonly used and seems safe without severe toxicity. With a mean interval of well over one year, the mean decline in eGFR in our study was only 6.0 (SD ± 9.8) mL/min/1.73 m^2^. This value corresponds well with the mean decline in eGFR of 5.5 (SD ± 13.3) mL/min/1.73 m^2^ that was described in previous SABR studies [15,28]. This limited decline in renal function in our patients with relatively large RCC may well be the result of this gated approach with small mobility boundaries, instead of using internal target volumes incorporating full tumor motion.

MRgRT also offers the advantage of using plan re-optimization for each delivered fraction at the cost of additional time. Our offline analysis showed that daily plan re-optimization was required in only 16% of fractions in which the predicted plan failed to meet the predetermined high-dose OAR constraints or target coverage objectives. Decision tree analysis showed that patients for whom daily plan re-optimization is not required can be identified upfront on the basis of a V_25Gy_ of the combined OAR of less than 0.5 cc in the baseline plan. It is, however, unlikely that an isolated single fraction violating high OAR dose or target constraints will be clinically relevant, and three out of five insufficient predicted plans were seen in only 14% of patients. Performing MRgRT without plan re-optimization indicates that the re-contouring, plan adaptation, and plan quality assurance phases can be omitted, which would enable respiratory-gated MRgRT fractions to be completed in 30 min. Furthermore, when plan adaptation is redundant, this indicates that the presence of the radiation oncologist at the MR Linac is not necessary. As a result of our analysis, we are currently introducing the found V_25Gy_ selection criterion in clinical practice.

The main limitation of our study is the relative short and unstructured patient follow-up. The limited number of RCC patients reflects the limited role of SABR in current international treatment guidelines, as only patients unsuitable for or refusing other local treatments are referred for curative radiation therapy. Another limitation includes the absence of pathology in half of our patients. Incomplete pathology confirmation is partly inherent to our patient population with generally frail elderly patients, which is unsuitable for other treatment modalities. Moreover, in a number of patients, a diagnostic biopsy was considered contra-indicated because of anticoagulant use or the anatomical location of the tumor. All patients had been discussed in a multidisciplinary tumor board with access to all available diagnostic imaging. Contrast enhanced multi-phasic CT has a high sensitivity and specificity for characterization and detection of RCC [3,29] and this specific imaging was available for all patients without pathological confirmation.

Prior to the MRgRT era, the need for radiologists to implant fiducial markers has also been an obstacle for referral for SABR. Our data show that MRgRT can be a valid alternative in patients unsuitable for the more commonly used local treatments, because of patient vitality or tumor size. The only contra-indication for MRgRT is having MR-incompatible devices. The main advantage of MRgRT is that it is an outpatient, non-invasive treatment for which not even the placement of fiducial markers is necessary. Whether MRgRT can also be considered as an alternative to partial nephrectomy or cryotherapy needs to be addressed in a prospective randomized study, which should also evaluate quality of life and cost-effectiveness. With regard to the favorable outcome in the data on SABR literature as well as the current analysis on MRgRT, a more prominent role of SABR in the treatment guidelines for RCC appears warranted.

## 5. Conclusions

In conclusion, hypo-fractionated MRgRT for large RCC resulted in high LC and very low toxicity rates. Gated treatment without the need for anesthesia or fiducials appeared well tolerated. Even in this group with large RCCs, daily plan re-optimization was not needed for the majority of patients, who can be identified upfront by a combined OAR V_25Gy_ of ≤ 0.5 cc in the baseline plans. This is a favorable result since online MRgRT plan adaptation is a time-consuming procedure. In this group of patients, MRgRT delivery will be faster, and these patients could be candidates for further hypofractionation [30].

## Figures and Tables

**Figure 1 cancers-12-02763-f001:**
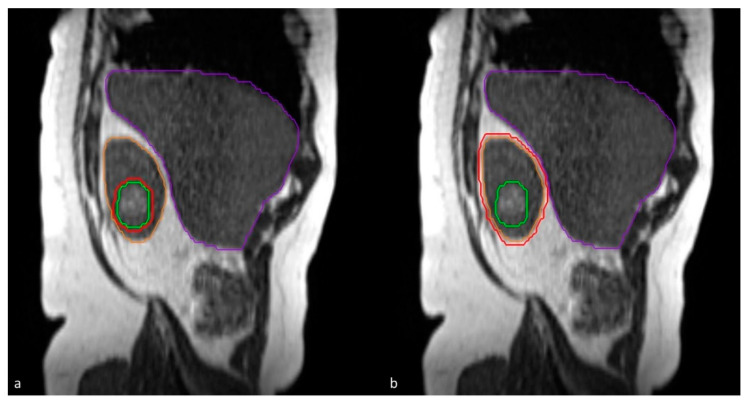
Sagittal plane for tumor tracking: either (**a**) tracking on gross tumor volume (green) or (**b**) tracking on the whole kidney (orange). A boundary of 3 mm (red) for gated delivery.

**Figure 2 cancers-12-02763-f002:**
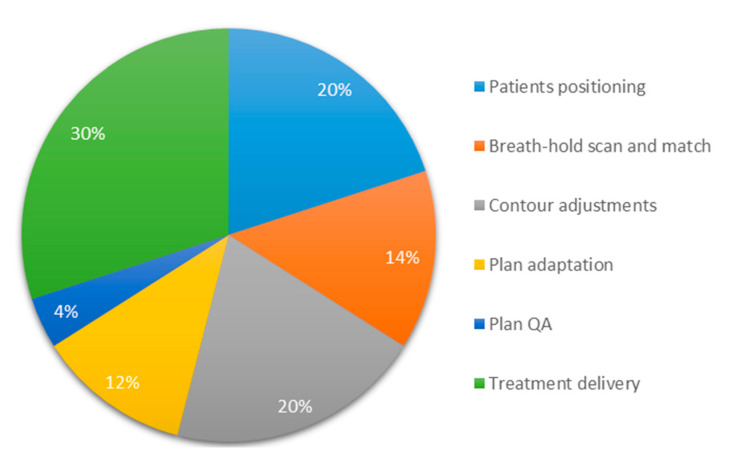
Pie-chart of the average duration of the different components of breath-hold gated adaptive MR-guided radiotherapy with an average time per fraction of 45 min.

**Figure 3 cancers-12-02763-f003:**
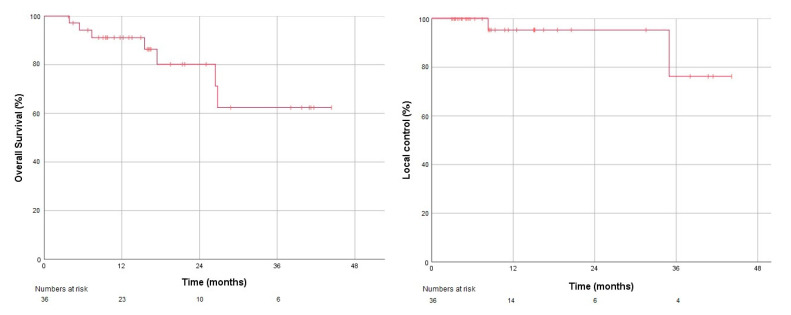
Kaplan-Meier plots for overall survival (left) and local control (right).

**Figure 4 cancers-12-02763-f004:**
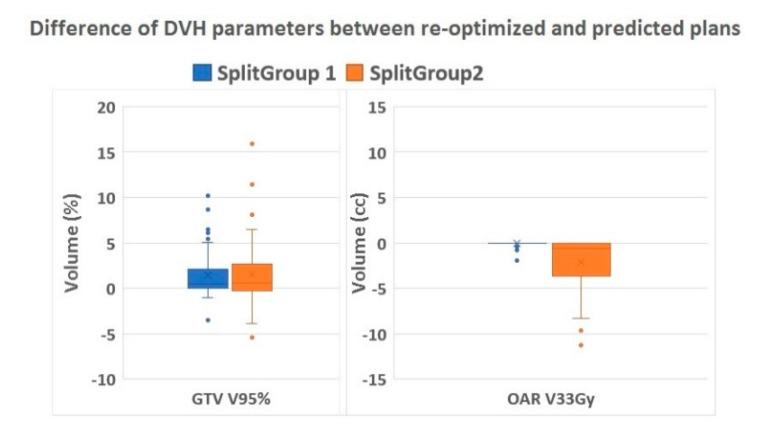
Difference of DVH parameters. Boxplots showing the relative volume difference in GTV V_95%_ (%) and absolute difference in OAR V_33Gy_ (cc) of the re-optimized compared to the predicted plans stratified for Split group 1 (re-optimization not needed) and 2 (re-optimization needed). Abbreviations: DVH = dose volume histogram, GTV = gross target volume, OAR = organs at risk.

**Table 1 cancers-12-02763-t001:** Dose prescription for institutional target coverage and normal tissue constraints. The constraints represent the cut-off doses for radiotherapy planning with the aim of dose sparing in the surrounding organs (contralateral kidney, liver, duodenum, bowel, and stomach) while, at the same time, aiming to achieve a high dose in the tumor with the margin, which is represented as the planning target volume. Organs at risk are only re-contoured within 2 cm of the tumor and, for an adaptive setting, only the dose in these structures are optimized.

Structure	Dose to Volume			
Planning Target Volume	≥50	% at	38	Gy
	≤1	cc at	50	Gy
Kidney Contralateral	≤25	% at	12	Gy
Liver	≤50	% at	12	Gy
Duodenum, Bowel, Stomach in 2 cm	≤0.1	cc at	36	Gy
	≤1	cc at	33	Gy

**Table 2 cancers-12-02763-t002:** Baseline patient characteristics (*n* = 36). Abbreviations: RCC = renal cell carcinoma, GTV = gross tumor volume, PTV = planning target volume, CKD = chronic kidney disease.

Mean Age (Range), Years		78.1 (58–95)
Sex, *n* (%)		
Male		24 (66.7)
Female		12 (33.3)
WHO performance status, *n* (%)		
0		3 (7.9)
1		21 (58.3)
2		12 (33.3)
Charlson comorbidity, *n* (%)		
Mean (SD)		6.4 (2.5)
2–3		3 (8.3)
4–6		18 (50)
7–9		10 (27.8)
10–13		5 (13.9)
Histology RCC, *n* (%)		
Yes		20 (55.6)
No		16 (44.4)
Tumor Laterality, *n* (%)		
Left		13 (36.1)
Right		23 (63.9)
Tumor location, *n* (%)		
Interpolar		13 (36.1)
Lower pole		13 (36.1)
Upper pole		10 (27.8)
Tumor size largest dimension, cm		
Mean (SD)		5.6 (1.6)
Median (range)		5.5 (2.4–9.3)
T-stage, *n* (%)		
cT1a		5 (13.9)
cT1b		23 (63.9)
cT2a		8 (22.2)
GTV, cc		
Mean (range)		79.7 (7.7–350.4)
PTV, cc		
Mean (range)		108.6 (14.3–445.9)
Renal function (eGFR), ml/min/1.73 m^2^		
Mean (SD)		55.8 (20.1)
CKD classification, *n* (%)		
I	Normal (eGFR ≥ 90)	0 (0)
II	Mild (eGFR ≥ 60 to < 90)	15 (41.7)
IIIa	Mild-Moderate (eGFR ≥ 45 to <60)	10 (27.8)
IIIb	Moderate-Severe (eGFR ≥ 30 to <45)	8 (22.2)
IV	Severe (eGFR < 30)	2 (5.6)
V	Kidney failure (eGFR < 15)	1 (2.8)

**Table 3 cancers-12-02763-t003:** Results in the Chi-square automatic interaction detection (CHAID) tree table.

	Redundant *n* (%)	Needed *n* (%)	Total *n* (%)	Predictive Variable	Split Values	Chi-Square	df	*p*-Value
Parent node: all cases	151 (83.9)	29 (16.1)	180 (100)					
Split group 1	90 (100)	0 (0)	90 (100)	OAR V_25Gy_	≤0.5 cc	34.6	1	<0.001
Split group 2	61 (67.8)	25 (32.2)	90 (100)	OAR V_25Gy_	>0.5 cc	34.6	1	<0.001

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
