# Peer review of "The Role of Daily Adaptive Stereotactic MR-Guided Radiotherapy for Renal Cell Cancer"

_cancers, 2020, doi:10.3390/cancers12102763_

Round 1

Reviewer 1 Report

This is an interesting paper about a novel modality of local therapy for renal cell carcinoma. Some minor comments:

  • Page 2, line 58: "Organs at risk" abbreviation should be defined here rather than in line 76: "organs at risk (OAR)".
  • Page 2: Table 1 may be difficult to understand by non-radiation oncologist experts. Please consider to better explain meaning.
  • Page 4, lines 145-147: The proportion of patients undergoing biopsy or pathologic confirmation of renal cell carcinoma is somehow low in my view. Although authors comment possible reasons, please consider to reflect more specifically these ones.
  • Local control is 96.4% and overall survival is 90.8%. To me, it is somehow odd. I understand that 96.4% refers to 34 out ot 36 patients that could be evaluated. I think this point could be better remarked.

No other concers to me. Thank you.

Reviewer 2 Report

This article is narrative stereotactic MRgRT for large primary RCC showed low toxicity and high LC, while daily plan re-optimisation was required only in a minority of patients.The content is simple and concise , it I feasible result. But there are some problem please explain.

  1. Why the half of the RCC patients in the table have no pathological proven.
  2. It is another option for renal tumor treatment, but what is the difference between it and partial nephrectomy ,cryotherapy.
  3. How much is the degree of tumor shrinkage after treatment, is there a difference between the left and the right.
  4. what percentage of MRRT need to have hospitalization and what percentage need surgery after MRgRT
  5. The toxicity of radiotherapy will slowly grow after two years. Apply to ask how many patients have been tracked for more than two years.
  6. what percentage of these MRgRT patients use target or immunotherapy at the same time.
  7. what is the patients quality of life after MRgRT and cost effectiveness.
  8. Please add the tumor grading , what type of data stores are there?w

Reviewer 3 Report

In this retrospective study, Tetar et al. describes their institutional experience of MRgRT for RCC. Only minor points are raised.

Minor points:

1) Changes/edits continue to be tracked. Please remove.

2) Please include more details of treatment planning, including the machine used for MRgRT, planning treatment software, homogeneity correction, software and algorithm used for deformable image registration

3) Including DVH's of target and OAR's with and without re-planning, would strengthen the conclusions

4) In figure 3, what is the difference between the top two Kaplan-Meier curves and the bottom two?

Reviewer 4 Report

Dear Dr. Bruynzeel et al

This paper describes the utilisation of MR guided radiotherapy deliver for the treatment of cancers that reduces clinical time and increases accuracy of delivery. This is a very important topic and demonstrates the need to shift to MRI guided therapies. Overall the manuscript is well written however there are issues 

Major comments

The authors have relied heavy on the high frame rate and geometric accuracy of MRI. Please address the below points

  1. The sequence parameters & coils used and MRI system details must be described in full. The B0, RF and duty cycle has a big impact on image quality. Please add a full description? 
  2. What is the geometric accuracy of your system and how did you measure this? if you are drawing margins this is vital in determining the accuracy of therapy. 
  3. Please address any limitations of using MRI in the conclusions

Minor comments

Abstract line  line 26

Introduction 37-48

and others 

It seems that the wrong PDF may have been uploaded with errors that must be corrected. 

Round 2

Reviewer 2 Report

The tissue pathology report is still very important, because th results of treatment are related to him, please provide detailed information

Reviewer 4 Report

NA

Round 3

Reviewer 2 Report

The pathological report is very important for cancer control.If the author has no way to know the detail path logic report clearly, it is difficult to know the real prognosis and survival rate of the disease.